# A case study of the spectral parameters of ULF fluctuations before substorms with no evident trigger in the interplanetary space

Nataliya Sergeevna Nosikova[1,2], Nadezda Viktorovna Yagova[2], Lisa Jane Baddeley[3], Dag Arne Lorentzen[3], Dmitriy Anatolyevich Sormakov[4]

[1]National Nuclear University "MEPhI", Moscow, Russia
[2] Schmidt Institute of Physics of the Earth of the Russian Academy of Sciences (IPE RAS), Moscow, Russia
[3]University Center on Svalbard, Norway
[4]Arctic and Antarctic Research Institute, AARI, Geophysics department, Sankt-Petersburg, Russia

*Correspondence to*: Nataliya S Nosikova (NSNosikova@mephi.ru)

**Abstract.**

Our recent study (Yagova et al, 2017) shows statistically that ultralow frequency (ULF) pulsations are seen in ground-based magnetic and luminosity data in the polar cap for a 3 – 4 hours preceding the onset time for isolated, non-triggered substorms. Such pulsations are characterized by a high level of coherence. It was discussed whether the pulsations represented a signature of a substorm preparation phase of the magnetosphere and if so, could the pulsations be attributed to an external source in the solar wind or to processes inside the magnetosphere. To investigate this further a detailed case study of an isolated non-triggered substorm has been carried out using co-ordinated ground and space based instrumentation. Fluctuations at ~1.5 mHz are observed in the both polar caps in ground-based geomagnetic data and also in electron concentration in the northern hemisphere ionosphere. Coherent pulsations with a relatively narrow Pc-like spectra and a higher fraction of the transversal components in the total spectral power are also observed by the cluster satellites in the magnetotail magnetic field. Interestingly, the pulsations in the magnetotail started after pulsations over a similar frequency range observed in the solar wind dynamic pressure and IMF had been switched off. It is possible that such coherent pulsations (which are normally masked by a higher amplitude broadband ULF "noise" of extra-magnetospheric origin during traditional substorms) do represent a substorm preparatory phase of the magnetosphere several hours before substorm onset.

## 1 Introduction

Substorms are a spectacular phenomenon, which occur in the near-Earth space, which have been studied for more than 50 years. A substorm is essentially a rapid energy release in the magnetotail, which is recorded as a local change in the geomagnetic field in the night sector at high latitudes and is accompanied by auroral luminosity brightening. The energy supplied from the Solar Wind (SW) into the Magnetosphere might immediately be unload or be accumulated until it exceeds a defined limit (Akasofu, 2017). This process is traditionally counted as a first part of a substorm and called a growth or loading

phase. The amount of energy entering to the Magnetosphere can be estimated using the Akasofu parameter ($\varepsilon$). A medium substorm can develop, if the hourly average value of $\varepsilon = 3 - 5 \cdot 10^{11}$ J/s. If $\varepsilon$ is slightly above $10^{11}$ J/s, it might take up to 3-4 hours for a magnetospheric response. Liu et al. (2013) reported of nine-hour-long growth phase. The following unloading processes (the expansion phase) might be triggered by changes in the SW and Interplanetary Magnetic Field (IMF) or some magnetotail events (internal triggers). Hsu and McPherron (2009) gave a detailed description of substorm triggering studies since 1955. The authors summarised previous papers and identified 3 main possible external triggers: «northward turnings of $B_Z$, reductions in IMF $|B_Y|$, and changes in dynamic pressure». Newell et al. (2016) argued that changes in the speed of SW increase onset probability. The authors noted that they were interested in external conditions, which lead to a substorm, rather than «investigating the immediate trigger». According to (Akasofu, 2017), substorm triggering is one of the controversial topics.

Another important energy transfer mechanism is through the propagation of ULF (Ultra Low Frequency) waves. Particularly, intensive high latitude geomagnetic pulsations at mHz frequencies (Pc5/Pi3) are of key interest. These pulsations are the most long-period and intensive magneto-hydrodynamic (MHD) waves in the magnetosphere. Their longest possible period is determined by the magnetosphere size to Alfven velocity ratio and is about $10^3$ s. Intensive Pc5 pulsations result from the Alfven field-line resonance (FLR) on closed field lines. Magnetospheric global pulsations have been linked to transient or quasi-periodic fluctuations of the solar wind pressure (density) and IMF. Kepko et al. (2002) suggested, that the primary cause of such pulsations is ULF perturbations in the pressure and the density of the SW. Later, Kepko et al. (2020) carried out a detailed analysis of long-term observations of quasi-periodical spatial mesoscale irregularities in the SW number density. The authors have shown that at least some of the frequencies of global pulsations in mHz range can originate from the meso-scale SW structures. If the ULF variations in the SW are neglectable, pulsations might result from exciting Kelvin-Helmholtz instability in the magnetosphere (Mann et al., 2002, Rae et al., 2005, Keiling 2009).

Inside the magnetosphere, the spectral parameters of ULF waves are determined by spectral characteristics of disturbances outside the magnetosphere and the waveguide and resonance properties of the magnetosphere-ionosphere system (e.g., Alperovich and Fedorov, 2007).

Samson et al. (1992) suggested that FLRs can play a role in substorm triggering This idea was developed by Rae et al. (2014) who studied resonance ULF waves with high azimuthal wave numbers and concluded them as "a strong candidate for triggering substorm onset". Pi1/Pi2 fluctuations are always observed around substorm onset on the ground and in space. An exponential growth of 10-100 s ULF amplitudes in the magnetotail (Smith et al., 2020) was found simultaneously with the increase of auroral brightness. The problem of localization and timing of an onset and ULF activity during last pre-onset minutes has been studied rather intensively (see e. g. Smith et al., 2020 and references therein), while the role of ULF activity in the magnetotail at the early stages of substorm preparation (i.e., several hours before substorm onset) has not yet studied in detail. Nevertheless, variations of ULF and auroral activity has been reported in several studies (see Yagova et al., 2017 and references therein).

Pc5/Pi3 are also observed in the polar caps, where field lines are "open". The oscillations are characterized by lower amplitudes as compared with auroral Pc5s, they are more irregular with typically lower central frequencies ~ a few mHz (Bland, 2016).

These pulsations are simultaneously seen in geomagnetic and radar data and might be directly driven by perturbations in the SW, be related to substorms or be a result of internal processes in the magnetosphere (see references from introduction by Bland, 2016). A contribution of the main SW/IMF pulsations to the spectral power of polar cap Pi3s was estimated by Yagova et al. (2007). However, ULF spectra at polar cap latitudes are not totally controlled by the parameters in front of the bow shock but is a manifestation of processes in the magnetosphere and magnetosheath (see e.g., Yagova, 2015 and references therein).

Pilipenko et al. (2005) theoretically showed, that an Alfven quasi-resonator can be formed on open field lines due to the curvature of the magnetic field. This mechanism can provide an explanation ULF activity observed in the polar cap. Due to a large spatial scale of these pulsations, their total energy is essential, and their amplitude correlates with the flux of accelerated electrons in the magnetosphere (Nasi et al., 2020). In view of this, the ULF activity can be important in substorm preparation. This manuscript is based on our previous paper (Yagova et al., 2017) and it develops the idea proposed in the paper. The

mentioned study is aimed on the investigation of geomagnetic and auroral luminosity pulsations in the frequency range 1-4 mHz using ground-based magnetometer and Meridional Scanning Photometer data. Days with undisturbed SW and IMF parameters were selected and divided into 2 sets: days with a substorm and quiet days. The second set was used to determine a background variation of the spectral parameters of the ULF pulsations. The first set consists of 15 non-triggered substorms, i.e., substorms without any evident trigger in the IMF nor SW; 7 out of 15 substorms were considered as isolated, i.e., separated

from other substorm by at least 3 hours. It was shown, that for days with non-triggered substorms, the Power Spectral Density (PSD) of Pc5/Pi3 geomagnetic pulsations, recorded in the Northern Polar Cap for several hours, preceding a substorm onset, is much higher as compared with the quiet days. The pre-substorm geomagnetic pulsations were coherent along a magnetic meridian and with fluctuations in auroral luminosity (557.7 nm). The analysis also showed distinct changes in the spectral content of the pulsations. The most important pre-substorm signature was the occurrence of pulsations with a clear spectral

maximum, i.e., their quality parameter $Q$ is higher than that for typical for polar cap Pi3s. These changes were observed during 3-4 hours before a substorm onset. It was speculated that these highly coherent pulsations, inside the polar cap might be directly caused by ULF activity in the IMF and SW or be a more indirect signature of a preparatory phase of the magnetosphere to an upcoming substorm.

The present case study illustrates the possibility of the 'preparatory phase' scenario by combining ground magnetometer with

90 in-situ measurements inside the magnetosphere (from Cluster). The analysis is thus focused on the time period, 3-4 hours before substorm onset, in line with the finding from Yagova et al. (2017). While ULF fluctuations almost die out in the SW and IMF, ULF activity starts to grow and become more coherent in the magnetosphere as shown from both the Cluster and ground magnetometer data.

The present study is focused on an isolated, non-triggered substorm (on 08.08.2007); which was recorded at an auroral

magnetometer station. The Dst index for the previous 4 days indicated quiet geomagnetic activity with a maximum value of - 50nT. For 6 hours preceding the substorm, the SW and IMF stayed moderate and undisturbed (see Figure 2 and section 3.1 for details). The studied substorm can be classed as isolated, since the maximum value of AE did not exceed 80 nT for at least 3 previous hours. The substorm perfectly fits in the dataset of selected days used in the paper (Yagova et al., 2017).

## 2 Data processing

### 2.1 Data

Data from the IMAGE magnetometer network, VOS, DVS, DRV, THL magnetic stations, EISCAT radar, CLUSTER, DMSP as well as OMNI Solar wind and IMF data were utilized. The position of the Cluster satellites and the field line projection of their position (between 12 and 18 UT) together with the location of the stations are given in Figure 1. It should be noted, that the both satellites were in the tail Southern tail lobe for the full day (not shown).

IMAGE is a European magnetometer network equipped with three-component flux-gate magnetometers with 10 s initial time resolution (Tanskanen, 2009). The chain is located approximately along the magnetic meridian 100 (MM100), and it covers CGM latitudes from $\Phi=77°$ to $40°$. Information on the stations utilised in this study and their coordinates is presented in Table 1.

The EISCAT Svalbard radar (ESR) is co-located with the LYR magnetometer station ($\Phi=75.4°$) and can make detailed measurements of ionospheric electron density, electron temperature, ion temperature and ion line-of-sight velocity. The radar operates in the 500 MHz band with a peak transmitter power of 1000 kW. In the present paper, data from the 42-meter dish (field aligned pointing position) taken during the International Polar Year (2007–2008) are used.

To map the magnetic field variations from the magnetotail to the ground, data from Cluster satellites fluxgate magnetometer have been included (Balogh et al., 1997). The values of footprint coordinates are taken from https://sscweb.gsfc.nasa.gov/cgi-bin/Locator.cgi. The Tsyganenko 89C model is used for field line tracing

On the $8^{th}$ of August 2007 (day 220) all four Cluster satellites were located in the southern tail lobe at radial distances of between 15 to 20 Re. Specifically, fluxgate magnetometer data from Cluster 1 and 3 are used (which had a separation distance of approximately 1 $R_E$ along the Sun-Earth line). Using the TS01 model, Cluster 3 was nearly conjugated with the DRV station (its southern footprint at 17:10 UT was at -68.7°, 141.4° geographic and -76°, 236° CGM).

### 2.2 Data Processing

Preliminary data processing includes band-pass filtration in the frequency band 0.8-8.3 mHz and decimation to a common 1-minute temporal resolution. The lower frequency of the filter window corresponds to ~ 20 min period. An example of spectra of a non-filtered signal can be found in the Figure S1 in the supplementary materials. All spectral maxima at frequencies above 1.1 mHz are separated from the filter-related maxima in all the spectra shown as examples and used in statistical analysis.

In the case of the magnetospheric satellite measurements of the magnetic field, a local field-aligned system is used. For each time instant, there are two magnetic field vectors in GSE coordinate system, i.e., the instantaneous vector **B** and the averaged over the time window one **B**$_{av}$. The field aligned component $B_{||}$ is defined as a projection of **B** to **B**$_{av}$. The radial component $B_\rho$ is normal to **B**$_{av}$ and lies in the plane containing **B** and the Earth centre and directed downward. The azimuthal component $B_\phi$ is normal to both $B_{||}$ and $B_\rho$: its direction is determined from the condition that three components form a right-hand triangle.

Electron concentration registered by the EISCAT radar are preliminarily sliced with height, averaged, and passed through the

low-band filter with a cutoff frequency $f_c$=8.3 mHz. On the day of the case study, 8th August, Svalbard was experiencing the midnight Sun period so the level of background solar radiation incident on the ionosphere was high. Nevertheless, variations of electron density ($\Delta N_e$) are suitable for analysis.

The processing of all data (ground and space based) to a common temporal resolution allows a cross-spectral analysis to be performed between the various data sets. The Blackman-Tukey method (Jenkins, 1969; Kay, 1988) is applied to obtain a power spectral density (PSD) for each variable, along with PSD ratio ($R$), spectral coherence ($\gamma^2$) and phase difference ($\Delta\varphi$) for each pair of variables. The spectra are calculated in a 96-points (5760 s) sliding window with an 8-min shift between subsequent intervals. Whilst the Blackmann-Tukey method has a more course frequency resolution than other similar methods (such as the Maximum Entropy method), it estimates the PSD with a dispersion which decreases with spectral smoothing. The spectral estimation parameters were chosen as a compromise between two opposite requirements, namely a better frequency resolution and a lower dispersion of the spectral estimation. It should be remembered that the dispersion of spectral coherence and phase difference depends on the absolute value of coherence and goes to zero at $\gamma^2 \rightarrow 1$.

OMNI data (time delayed to the Bow Shock) are used for spectral estimations in the IMF whilst Cluster data are used for the magnetotail. Various studies have looked at the response time of the magnetosphere to changes in the Solar Wind (primarily to changes in the IMF). Wing et al. (2002) found a the nightside response time to changes in the IMF of ~12 minutes (at geosynchronous orbit). Given that this study is looking at pulsations with a periodicity of ~12 minutes (1.5 mHz) and using a 1.5-hour sampling window for PSD and coherence estimates then the OMNI data has not undergone any additional time lag when evaluating the coherence between that and the Cluster data.

## 3 Results of the analysis

### 3.1 Non-triggered substorm on day 220 2007 and space weather conditions

A weak substorm, registered at 20:30 UT on 8th of August 2007 (Day 220) at TRO station is studied in detail. The magnetic field data from the TRO and BJN stations and AE-index, as well as IMF and SW parameters are shown in Figure 2, left panel. It is seen from the Figure, that the IMF was undisturbed, indicating no features which could be considered as a substorm trigger, and the absolute value of the $B_Z$ component hardly exceed 3 nT. The absolute value of the solar wind speed $V$ was slightly decreasing for the entire period and the mean value was about 500 km/s, which corresponds to moderate SW. The SW dynamic pressure stayed low and there were no rapid changes, which can be considered as an external substorm trigger. The AE index was increasing from 14 UT to 16 UT and there were two enhancements: around 17 UT, and at 20:30 UT. The second enhancement, at 20:30 UT, marked with the vertical blue arrow, corresponds to the substorm studied. Although the magnitude of the substorm bay is small (~100 nT), observations by DMSP-satellite (F16, Figure 2b, top right panel) indicate auroral enhancements consistent with a substorm (i.e., particle precipitation occurring from the magnetotail). It should be noted that even if the first enhancement at 17 UT relates to a substorm activation, it does not break the criteria required for the second

substorm to be considered isolated. It also should be added that there is no ground based optical observations available in the high latitude northern region due to 24-hour daylight conditions in northern Scandinavia.

## 3.2 ULF fluctuations on the ground and in space

In this section, the analysis of ULF fluctuations in interplanetary space, the magnetotail and on the ground are presented. The parameters of the IMF/SW fluctuations and geomagnetic pulsations in the magnetotail are calculated throughout the day to illustrate how the parameters of the pulsations in the magnetotail and on the ground are controlled by extra-magnetospheric parameters and the changing (i.e., moving from the auroral oval to the polar cap) of the magnetospheric projection of ground stations. Such a time scale is used in Figures 4(a),6, 8, and 10. For a more precise analysis of the pulsation spectral properties
during a few pre-substorm hours, a shorter time interval is used in Figures 4(b, c), 11, 12, and 14.

### 3.2.1 Geomagnetic pulsations on the ground: Northern hemisphere

As was mentioned above, it is expected to find Pc5/Pi3 pulsations, similar to pre-substorm pulsations (Yagova et al., 2017), in the polar cap before the studied substorm. Indeed, ULF pulsations are observed in the ground magnetometer data stations located on Svalbard (from NAL to HOR) and at BJN and become indistinct at TRO station, situated on the mainland (Figure 3).
For the ground stations, the notations $B_N$ and $B_E$ are used for the components oriented northward along the magnetic meridian, and eastward (orthogonal), respectively. The time series starts at 15:00 UT and stretches for 180 minutes. The maximal amplitude reaches ~25 nT peak to peak at ~16:30 UT for LYR and NAL and then decreases for the value of a few nT. The diurnal variations of the Pi3 (1-4 mHz) power at NAL is controlled by two active regions: the polar cusp, which is responsible for near-noon activity, and the polar boundary of auroral oval in the morning and evening MLT sectors (see Yagova
et al. (2004) for details). The station crosses both zones at different UTs with a dependence on seasons and geomagnetic activity in the morning and afternoon MLT sectors. The PSD variation during the day analysed is shown in Figure 4(a). Each data point along the time axis in Figure 4 (b) corresponds to a starting point of a nearly 1.5-hour (96 minute) interval. After the cusp-related maximum near MLT noon (UT=9) (indicated by the vertical red arrow), the PSD reaches a maximum at 13 UT (16 MLT) (indicated by the vertical purple arrow). Afternoon Pi3s, seen after 15 UT, do not relate to the cusp activity and
are the object of our special interest. To clarify how the contribution of polar cap and auroral oval activity to the NAL pulsations changes with time, the variation of the Pi3 PSD ratio, $R$, and spectral coherence, $\gamma^2$, for THL-NAL and HOR-NAL station pairs are given on right (Figure 4, b-c). HOR is located ~ 3 degrees southward from NAL on the MM100 chain, while THL lies at $\Phi=84.84°$, i.e., deep in the polar cap. Although THL is shifted by 5.5 hours in MLT from the MM110 stations, the diurnal variation is almost negligible
for this location (Yagova et al., 2010) and this station can be taken as an indicator of condition in the polar cap. As it was shown (Yagova et al., 2017), the main pulsation power is concentrated in the frequency band from 1.25 to 1.9 mHz, and UT from 8 to 20, i.e., from pre-noon to almost midnight in MLT. It is seen from the Figure 4b, that both the THL/NAL and HOR/NAL PSD ratio, $R < 1$, at near-noon hours, i.e., NAL is dominating, probably due to cusp-related activity. From 11 to

16:30 UT, the THL/NAL spectral ratio, $R > 1$, while HOR/NAL spectral ratio, $R \sim 1$, i.e., the PSD deep in the polar cap is maximal. The HOR/NAL PSD ratio then grows while the THL-NAL PSD ratio remains at $\sim 1$. From 13 to 17 UT, the pulsations are coherent for both the THL-NAL and HOR-NAL station pairs with coherence maxima at 14 and 16 UT (Figure 4c). For the first maximum, the HOR-NAL spectral coherence is higher, while for the second one the coherence for the THL-NAL station pair exceeds that for HOR-NAL. This means that after 15 UT, the polar cap (THL) pulsations demonstrate both the highest PSD and coherence with those at NAL. This effect can be associated with NAL moving from the auroral oval to the polar cap or can also result from temporal variations of the pulsation parameters, such as PSD distributions along a meridian and/or spectral coherence.

An example of the $B_N$ component variations and their spectral parameters are presented in Figure 5. The left panel (Figure 5a) shows a time series of the $B_N$ component from 15:44-17:20 UT from the NAL, THL and HOR stations. The pulsation has a period of approximately 11 minutes ($\sim 1.5$ mHz) and the peak-to-peak amplitude is about 25 nT at NAL and THL, and $\sim 15$ nT at HOR. Note, that the lower frequency of the filter $f_L = 0.8$ mHz corresponds to $\sim 20$ min period which is almost twice longer than the main pulsation's period. Figures 5b – 5d show the (b) PSD in each location, (c) the interstation spectral coherence, and (d) phase difference. The main maximum of the PSD at all the stations is found at $f_1 = 1.5$ mHz and the value of the PSD decreases with CGM latitude from THL to HOR (Figure 5b). This spectral coherence for both the NAL-THL and the NAL-HOR station pairs also demonstrate a maximum at $f = f_1$ (as is shown in Figure 5c). Note, that the spectral coherence between NAL and THL (green line in figures 5c and 5d) is higher, than between NAL and HOR (blue line in Figures 5c, d). The pulsations at NAL and THL are almost in phase in the vicinity of $f_1$, while a phase difference of $\sim \pi$ exists between NAL and HOR. Similarly, to the pre-substorm pulsations analysed in (Yagova et al., 2017), the pulsations in the polar cap are characterised by a clear spectral maximum, and the only difference from typical auroral Pc5s, is a lower frequency of the main maximum ($f_1 = 1.5$ mHz).

### 3.2.2 Magnetic pulsations in the magnetotail and interplanetary space

In (Yagova et al., 2017), two possibilities for the development of a substorm with no clear external trigger in non-wave parameters of the IMF or SW were discussed. Firstly, if there are coherent ULF pulsations observed both internally to the magnetosphere and in the SW or IMF then these could act as a direct (or alternative) trigger for substorms (in addition to those already discussed for traditional substorms). Another possibility is that, in the absence of coherent pulsations in the SW or IMF, pulsations inside the polar cap, in the hours leading up to substorm onset, could represent a preparatory phase of the magnetosphere for a substorm. In this phase, it is possible that the traditional external substorm trigger is not needed and a purely non-triggered substorm can occur. In this sense the trigger mechanism could be considered as a process which takes into account the conditions internal to the magnetosphere. To discriminate between these two possibilities, ULF disturbances in the SW, IMF, magnetotail and ionosphere were analysed. The average PSD and coherence of magnetic field fluctuations in the IMF, calculated from OMNI, and in the magnetotail, recorded by Clusters 1 and 3, are presented in Figure 6 (over the same frequency range, 1.2–1.9 mHz, as those presented in Figure 4). The variations in PSD of the three IMF components and field

aligned component $b_\parallel$ in the magnetotail (Cluster3) are given in Figure 6a. One can see from the Figure, that the spectral power in the IMF decreases rapidly at about 13.30 UT (black, dashed arrow in the Figure), and then at ~15 UT the decrease is seen in the magnetotail (black solid arrow in figure 6a). Figure 6(b) shows the corresponding PSD in the SW dynamic pressure, $P_{SW,}$ and density, $n$. Whilst there is an increase in spectral power from 10 – 15 UT, there is a sharp decrease at 15 UT, i.e., nearly simultaneously with that of the magnetic field in the magnetotail.

Simultaneously, the coherence between the pulsations, as measured by Cluster 3 and Cluster 1, in the magnetotail (Figure 6c) jumps almost to unity, while no severe changes in IMF-Cluster coherence occur. In the coherence panel (figure 6c), only the X-component of IMF is shown as the coherence variations between the Cluster field aligned component, and Y and Z IMF components are similar to the Cluster – IMF $b_X$ coherence. However, the latter demonstrates a closer agreement to the coherence variations within the magnetotail before the pulsation regime changed at about 15 UT (magenta curve in Figure 6c). A similar decrease is found in the coherence between the magnetic field in the magnetotail and $P_{SW,}$ fluctuations (Figure 6d). As the fluctuations of SW density and dynamic pressure were almost identical ($\gamma^2 \sim 1$), only the coherence variations between $P_{SW}$ and Cluster 3 are shown in Figure 6d.

The change of pulsation regime in the magnetotail is rather abrupt and it can be seen in the time domain as well. Figure 7 shows the pulsations in $b_\parallel$ registered simultaneously at Clusters 1 and 3 and their PSD spectra. The time series for the interval started at 15:04 UT is given in Figure 7a. The switch from more or less similar pulsations to almost identical ones is seen at about 15:30 and marked with an arrow. The PSD spectra for the interval started at 15:44 (Figure 7b) has the main spectral maximum at $f=1.5$ mHz, i.e., at the same frequency $f_1$, as observed by the ground magnetometers.

To quantitively describe the variation of the spectral shape of the magnetotail pulsations during the day we have used the method described by Yagova et al. (2010, 2015). The technique is based upon an expansion of the function $\sigma(F)$ (a log-log spectrum, where $\sigma$ and $F$ are the logarithms of PSD and frequency, respectively) into Ledgrendre polynomials with the resulting first 3 coefficients ($L_0$, $L_1$, $L_2$) providing the required quantitative description. In particular, the $Q$ parameter is used (where $Q=-L_2$), which estimates the deviation of the spectrum from an inverse power approximation (colour noise) near the central frequency of interest. Yagova et al., 2017, showed that the parameter $Q$ is higher for pre-substorm pulsations, than for the typical polar cap Pi3s.

The results for the $Q$-parameter analysis, along with the spectral power ratio and phase difference between the different magnetic field components in the magnetotail, as measured by Cluster 3 are shown in Figure 8. The $Q$ value increases rapidly for all 3 components ($b_\parallel$, $b_\varphi$ and $b_\rho$) after the PSD in the IMF decreases (i.e., after the switch off of the mHz fluctuations in the IMF (dash-dot arrow at about 13.30 UT) and it reaches maximal value at 15 UT. It should be noted that at 15 UT, the PSD in the SW pressure decreases rapidly, however, the $Q$ value for the $b_\parallel$ and $b_\rho$ magnetic field components in the magnetotail remains high ($Q > 0.5$) until 18 UT (and until 17 UT for the $b_\varphi$ component). For $b_\parallel$, $Q$ slowly fluctuates between -0.5 and 0.5 from 0 at 13 UT and afterwards reaches almost 1 within half an hour. The growth for the transversal components ($b_\varphi$ and $b_\rho$) after 13:30 UT is not as fast, and $Q$ reaches its maximal value after 15 UT.

It has been shown by Pilipenko et al. (2013) that the characteristics of the magnetotail are highly sensitive to the polarization parameters of the pulsations (in comparison to a simple amplitude based investigation). The wave polarisation characteristics were changing during the interval analysed, as seen from the variations of the spectral power ratio, $R$, and phase difference, $\Delta\varphi$, for the $b_{\parallel}$-$b_{\rho}$ and the $b_{\parallel}$-$b_{\varphi}$ component pairs (Figures 8b and 8c). For both pairs of components, $R$ exceeds unity almost all day long, i.e., the compressional component is dominating. The only exceptions are registered at 3, 8, and 15 UT. In the afternoon MLT sector, i.e., after 9 UT, it first increases from 2 to nearly 10 and then drops to 2 for the $b_{\parallel}$-$b_{\rho}$ and below unity for the $b_{\parallel}$-$b_{\varphi}$ component pairs. Between 15-18 UT the average $R$ value is 2-3 times lower than for the previous 3 hours. Averaged over the same frequency band, the sin of the inter-component phase difference, $\Delta\varphi$, is shown at the bottom panel. Before 15 UT, the parameter varies predominantly in the interval [-0.5, 0.8] and it differs for two component pairs. At 15 UT it changes to almost unity for both component pairs. This corresponds to $\pi/2$ phase difference between field aligned and each of transversal components, indicating a large-scale pulsation with a high fraction of transversal (Alfven) components. The interval of $\pi/2$ phase difference and low $R$ values are seen in Figure 8 from 15 till 18 UT, i.e., it coincides with the interval of low amplitude, high $Q$ and spectral coherence of Pi3 pulsations in the magnetotail.

### 3.2.3 ULF waves in the magnetotail and on the ground: Inter-hemispheric relationship

Since the Cluster satellites were in the southern tail lobe, only a footprint in the Southern polar region can be calculated. To understand, how the magnetotail pulsations are related to those recorded in the two polar cap ionospheres, the pulsations recorded at Cluster have been compared with those, recorded in the Southern Polar cap, and the latter with the Northern polar cap pulsations. A time series of the pulsations registered simultaneously in the magnetotail at Cluster 3 and in both polar cap ionospheres are presented in Figure 9 (a-c) and the spectral coherence for the pairs of components (1 ground, 1 magnetotail) are given in panels (d-f). Pi3s at the DRV station, which is nominally conjugated with Cluster-3 during this time period, have a peak-to-peak amplitude of about 2 nT (Figure 9a), and the maximal spectral coherence, $\gamma^2$, at 1.5 mHz ($f_1$) exceeds 0.9 for both field-aligned and transversal components (Figure 9d). It is the maximal value among all the satellite-ground pairs. Pulsations at the VOS station, located deep in the southern polar cap, are similar to the pulsations at Cluster-3 and at DRV. Their peak-to-peak amplitudes reach 4 nT (Figure 9b). The Cluster3-VOS spectral coherence for both components at VOS and the field-aligned component at Cluster-3 is shown in Figure 9e. It reaches a spectral coherence of 0.7 at the frequency, $f_1$ and is higher than the coherence between the pulsation at VOS and the transversal components at Cluster (not shown here). During this interval, the ground Pi3s are more intensive in the Northern than in the Southern hemisphere. Hence, the peak-to-peak amplitude at NAL is about 25 nT (Figure 9c). The maximal coherence, $\gamma^2 \approx 0.9$, between the NAL and Cluster pulsations is found for the $b_{\rho}$-$b_N$ component pair (Figure 9f). The results show that there is clearly a high coherence between the pulsations observed in the magnetotail and those in the polar cap ionospheres.

A time series of spectral power and coherence, in the 1.2 – 1.9 mHz frequency band, for different pairs of components for Cluster 3 and VOS is presented in Figure 10. The VOS station is taken, because it is located deep within the polar cap at any

local time, and thus the influence of the cusp and auroral activity is minimal. A decrease of spectral power at VOS starts immediately after the "switch off" of IMF fluctuations (this instant dash-dot arrow in Figure 10). However, the total decrease of spectral power in the ionosphere is not so severe as in the magnetotail. From 15UT (marked by the black solid arrow) the spectral power measured at VOS remains approximately constant, despite the fact that the spectral power at both Cluster and in the SW dynamic pressure ($P_{SW}$) (Figure 6b) has decreased significantly. As a result, the tail to ground (T-G) spectral power ratio $R_{T-G}$ during the interval 15-18 UT is high in comparison with the previous hours (Figure 10b). The spectral coherence is also higher than its average value during the day, especially for the $b_\parallel$ (Cluster)- $b_N$ (VOS) component pair (Figure 10c).

A time series of spectral coherence, in the 1.2 – 1.9 mHz frequency band, between Cluster 3 and 4 ground-based stations in the both hemispheres is shown in Figure 11. The DRV station is nominally conjugated to Cluster-3, the NAL station is located in the Northern hemisphere, at the substorm meridian, VOS and THL are placed deeper in the southern and northern polar caps respectively as it was mentioned above. The three columns of the Figure 11 correspond to the three magnetic field components in the magnetotail, and four rows to the four stations. The two ground horizontal magnetic components are colour coded with $b_N$ in green and $b_E$ in orange. The time interval from 8 to 20 UT corresponds to hours from local noon to midnight at the substorm meridian and includes the interval from 13 to 18 UT, i.e., from the switch-off of the IMF ULF activity until last pre-substorm hours. The highest coherence is found for the $b_N$ (DRV)-$b_\varphi$ (Cluster-3) component pair and last for 2 hours from 15 to 17 UT. During the interval, a large-scale pulsation with a high fraction of transversal (Alfven) components in the spectral power is recorded in the magnetotail. A high ($\gamma^2$=0.7) but short coherence maximum is seen also in $b_N$ (DRV)-$b_\parallel$ (Cluster-3) component pair. A similar to $b_N$ (DRV)-$b_\varphi$ (Cluster-3) time evolution is found for the $b_E$ (VOS)-$b_\parallel$ (Cluster-3) and $b_E$ (VOS)-$b_\varphi$ (Cluster-3) component pairs, but at somewhat lower absolute values of $\gamma^2$. As it is seen from the bottom two rows of panels, during the 15-17 UT time interval, the averaged coherence hardly exceeds 0.5 for all components Cluster 3 - NAL and Cluster 3 - THL, i.e., for stations, located in the Northern polar cap. Generally, the averaged coherence between magnetotail Pi3s with those observed in the northern polar cap is lower than that for the Southern polar cap, as is expected as the Cluster satellites are located in the Southern tail lobe.

The inter-relation between the pulsations in the polar caps cannot be completely described by their spectral coherence with the only a single location on open field lines, in the magnetotail. It seems possible to partly compensate for this with the analysis of the coherence between the two polar caps and within each cap. Since the ionospheric observations are available in the Northern hemisphere at the NAL longitude (see table 1), all possible pairs of horizontal components for two combinations of stations (VOS-DVS in the Southern hemisphere, and DVS-NAL between the two hemispheres) were analysed. It should be noted that the maximal coherence at high latitudes is possible not only for the corresponding components (Lepidi et al., 1996). Hence, a maximal coherence can be found for two polar cap stations not between both meridional components, but between e.g., meridional component at the first station and latitudinal component at the second station. The results for the time interval 8-20 UT are given in Figure 12. A high coherence ($\gamma^2$>0.5) is seen between the $b_N$ component at VOS and $b_E$ at DVS (Figure

12a) and between the $b_E$ component at VOS and both DVS horizontal components for the 15-18 UT interval (Figure 12b).
Inter-hemispheric coherence over the same time interval, maximizes for the $b_N$ components of DVS and NAL (Figure 12c).

The results of the coherence analysis between the pulsations in the magnetotail and those observed by ground magnetometers in both the Northern and Southern polar caps show that the Pi3 pulsation recorded after 15 UT are characterized by a high coherence both in a space to ground sense and also interhemispherically. A possible interpretation of this is summarized below:

1)  A compressional/shear Alfven wave in the magnetotail is propagating predominantly in transversal/field aligned directions, respectively. A high coherence between the pulsations observed by ground magnetometers in each polar cap demonstrate that these waves exist in both tail lobes.

       2)  This leads to coherent pulsations in both polar caps with a higher coherence between the meridional components for nominally conjugated positions and a higher cross-component coherence for the pulsations inside the Southern polar
cap.

### 3.2.4 Electron density fluctuations in the ionosphere

To examine electron density fluctuations in the ionosphere over different altitudes EISCAT radar data have been used. The background Ne level was high due to Solar Extreme Ultraviolet (EUV) ionisation but the application of a low-bound filter (see sect. 2.2) allows fluctuations of electron density $\Delta N_e$ for each altitude to be calculated.

A time series of $\Delta N_e$, centred in the F-region (at $h$=205 km) along with the $b_N$ component of magnetic field at NAL is shown in Figure 13a. The spectral power of both time series is shown in figure 13b, their spectral coherence in figure 13c and the phase difference in figure 13d. The analysis indicates a common spectral maximum at $f_1$=1.5 mHz (Figure 13b) with a wide coherence maximum with $\gamma_{max}\approx0.9$ (Figure 13c). The pulsations are in anti-phase with one another, as is clearly seen from both the time series and the phase-difference, which is nearly $\pi$ at the $f_1$ frequency (Figure 13d).

Figure 14(a) shows the temporal variation of the spectral coherence, $\gamma^2$ (in the frequency band 1.2-1.9 mHz) between $\Delta N_e$ and the $b_N$ component of the magnetic field over an altitude range covering the E and F-regions of the ionosphere (from 100 – 450km). Before 16:30 UT, the highest spectral coherence is registered with maximum at about 200 km. The same altitude of maximal coherence is seen from 16:40 until 17:30 UT. Several spots of high coherence were found at lower (~150 km centred at 17 UT) and higher (350 km around 16:30 UT and 420 km at 17-17:30 UT) altitudes. An altitude profile taken at 15:44 UT
is shown in Figure 14b. Spectral coherence is high ($\gamma^2 > =0.5$) in the altitude range from 120 to 350 km, and the maximal coherence is found at $h$ = 205 km. This high coherence between the geomagnetic and $N_e$ pulsations can be a result of modulated particle precipitation. The altitudes where the highest coherence is found, correspond to the penetration altitude of electrons with energies in the hundreds of eV.

### 3.2.5 Summary

To summarise the observational results:

- Pi3 geomagnetic pulsations with a frequency $f_1 \sim 1.5$ mHz were registered in both the Northern and Southern polar caps 3 – 4hours preceding onset time for a weak isolated non-triggered substorm on the 8th August 2007.
- These pulsations were also observed in the magnetotail with remarkably high quality and coherence both in space and between the magnetotail and polar cap in Earth.

- The pulsations are seen simultaneously in the ionospheric electron density enhancements in the Northern Polar Cap.

## 4 Discussion

The existence of a statistical relation between ULF pulsations in the polar cap and auroral activation was found by (Heacock and Chao, 1980, and Yagova et al, 2000.). However, these papers did not discriminate between the externally and internally triggered substorms. This step was done in the previous paper (Yagova et al, 2017), where the quasi-statistical analysis of ULF

pulsations in the polar cap was undertaken focusing on non-triggered substorms only (i.e., substorms with no obvious trigger in the SW). This analysis led to the conclusion of two possible reasons of the observed pre-substorm pulsations in the polar caps, the first related to an increased level of ULF fluctuations in the IMF and SW plasma parameters, and the other linked to instabilities internal to the magnetosphere.

An isolated, non-triggered substorm occurred on the 8th August 2007 at 20:30UT (as observed by the MM100 magnetometer

chain).  By its very definition, it occurred during a period of quiet SW conditions without any traditional substorm triggers. During the time period of 3 - 4 hours (15 – 18 UT) before substorm onset, highly coherent pulsations in the Pc5/Pi3 range were observed in the magnetotail and inside the polar caps.  A spectral analysis of the IMF and SW parameters indicated that the amplitude of pulsations of a similar frequency had decreased significantly by 13:30 and 15:00 UT respectively (Figure 6).

Observations made in the magnetotail (by the Cluster satellites) and in the polar caps (by ground magnetometers and the

EISCAT radar) indicate that, at the same time (15 UT), the Pc5/Pi3 ULF characteristics in this region changed. A most impressive feature of the pulsations in the magnetotail during the last pre-substorm hours is a high $Q$ factor, with a central frequency about 1.5 mHz and extremely high coherence between the two Cluster satellites. The visible pulsations are almost in-phase. At the same moment the contribution of $b_\rho$ and $b_\varphi$ (transverse) components to the total spectral power increases.  The pulsations are also recorded in both polar caps by ground magnetometers. A coherence analysis shows that the maximal

coherence is found for nominally conjugated positions in the magnetotail and in the southern polar cap ionosphere and between the two hemispheres for the transversal magnetic field components in the magnetotail and the $b_N$ in the ground magnetometer data.  For non-conjugated position in the same hemisphere the coherence is higher for the field-aligned component in the magnetotail and $b_E$ in the ground magnetometer data. This could mean that the wave is a combination of a compressional mode

and shear Alfven modes contributing predominantly to wave transport in transversal and parallel direction to **B**, respectively.

The pulsations also show a high coherence between variations of electron concentration, $\Delta N_e$, and the ground magnetic $b_N$ component in the northern polar cap ionosphere. The fact that it is registered in the electron concentration could indicate modulated particle precipitation into the ionosphere from the magnetotail.

These high quality Pc-like pulsations in the magnetotail probably correspond to some resonance magnetotail mode and started after the external fluctuations had been switched off. Thus, one can speculate that usually the pulsations in the magnetotail are

390 masked by a higher amplitude broadband ULF "noise" of extra-magnetospheric origin. The existence of quasi-resonance modes at open field lines has been discussed by previous authors, for different types of pulsations. Physically they are related to the reflection of inhomogeneities in the distribution of the Alfven velocity (e.g., Pilipenko et al., 2005). Also, as discussed by (Leonovich, and Kozlov, 2018) the laterally inhomogeneous structure of the nightside magnetosphere can also result in various resonance and waveguide MHD modes in the Pc3-6 frequency range.

The time needed to accumulate enough energy for a substorm should be longer for a reduced amount of power being supplied into the magnetosphere from the SW (as defined by the Akasofu parameter $\varepsilon$) and this would be the case for a non-triggered substorm. The case of developing of a substorm from the fluctuations inside the magnetosphere should be the slowest process and the timescale of several hours seems possible for such a scenario.

5.Conclusion

Data from this case study supports the idea outlined by (e.g., Kozyreva et al., 2007) that ULF pulsations in the Pc5/Pi3 frequency range, observed 3 – 4 hours before substorm onset, inside the polar cap region can represent a signature of a substorm preparation phase of the magnetosphere. Here we have focused on a so called non-triggered substorm and as future work we aim to test the following hypothesis in a statistical sense, using a larger data base:

1. Even if IMF and SW fluctuations at mHz frequencies is included in an analysis of possible external substorm triggers, a substorm can develop under IMF/SW parameters typical for non-disturbed days.

2. Pc5/Pi3 pulsations in the geomagnetic tail and in the polar caps might play some role in the development of a non-triggered substorm. They can be either an indicator (precursor), or an active agent in substorm preparation

3. Pre-substorm Pc5/Pi3 pulsations are characterised by relatively small amplitudes along with extremely large spatial

scale. They have some spectral features which are untypical for open field lines, i.e., relatively narrow Pc-like spectra and a higher fraction of the transversal components in the total spectral power. These pulsations are seen simultaneously in the magnetic field and in electron density in the ionosphere.

**Data availability:** Publicly available ground-based magnetometer data: IMAGE through https://space.fmi.fi/image/www/, INTERMAGNET through www.intermagnet.org. Magnetometer data from VOS station are available through (http://geophys.aari.ru) upon request. EISCAT data are publicly available through https://www.eiscat.se/scientist/data/. Cluster and OMNI data are publicly available through CDAWEB (https://cdaweb.gsfc.nasa.gov). The maps in Figure 1 are made with Natural Earth (Free vector and raster map data https://www.naturalearthdata.com).

**Author contribution**: NN suggested the event for the case-study, prepared ground-based data for preliminary analysis. NY developed software and performed data analysis. LB assisted with EISCAT data analysis and DL assisted with DMSP data analysis. DS provided Vostok data. The manuscript was prepared by NN and NY, after discussions with LB and DL.

**Acknowledgements:** The authors thank prof. V.A. Pilipenko for helpful discussions. The authors thank the institutes who maintain the IMAGE Magnetometer Array: Tromsø Geophysical Observatory of UiT the Arctic University of Norway (Norway), Finnish Meteorological Institute (Finland), Institute of Geophysics Polish Academy of Sciences (Poland), GFZ German Research Centre for Geosciences (Germany), Geological Survey of Sweden (Sweden), Swedish Institute of Space Physics (Sweden), Sodankylä Geophysical Observatory of the University of Oulu (Finland), and Polar Geophysical Institute (Russia). The results presented in this paper rely on data collected at magnetic observatories. The authors thank the national institutes that support them and INTERMAGNET for promoting high standards of magnetic observatory practice (www.intermagnet.org). The authors thank the Arctic and Antarctic Research Institute in Russia for providing ground magnetometer data from VOS station (http://geophys.aari.ru). The authors thank CDAWEB (https://cdaweb.gsfc.nasa.gov) for Cluster and OMNI data. EISCAT is an international association supported by research organisations in China (CRIRP), Finland (SA), Japan (NIPR and ISEE), Norway (NFR), Sweden (VR), and the United Kingdom (UKRI). This research was partly funded by the PolarProg and INTPART research programs under the Research Council of Norway (project numbers 246725 and 309135, NN, LB, DL) and by RFBR grant # 20-05-00787 A (NY).

The authors declare that they have no conflict of interest.

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

Table 1. Station coordinates and their parameters

| Station code | Geographic | | Geomagnetic (CGM) | | UT of MLT midnight |
|---|---|---|---|---|---|
| | Latitude | Longitude | Latitude | Longitude | |
| NAL | 78.92 | 11.95 | 76.34 | 110.45 | 20:59 |
| LYR | 78.20 | 15.82 | 75.40 | 111.20 | 20:55 |
| HOP | 76.5 | 25.1 | 73.22 | 114.53 | 20:40 |
| TRO | 69.66 | 18.95 | 66.75 | 102.42 | 21:26 |
| THL | 77.48 | 290.83 | 84.84 | 29.19 | 3:13 |
| DRV | -66.66 | 140.01 | -80.37 | 236.04 | 12:55 |
| VOS | -78.46 | 106.82 | -83.57 | 55.15 | 1:02 |
| DVS | -68.58 | 77.97 | -74.75 | 101.17 | 21:58 |

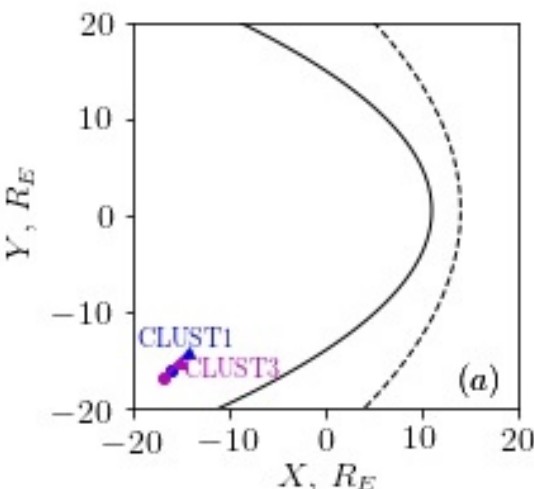
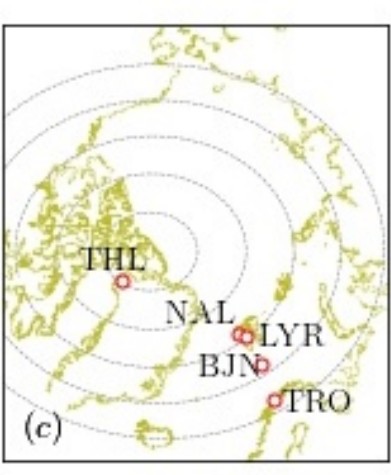
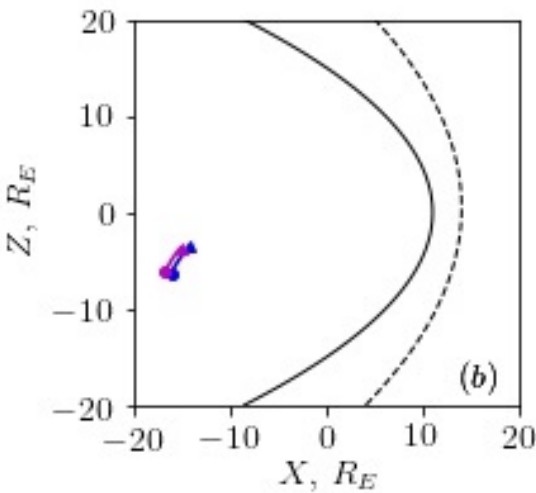
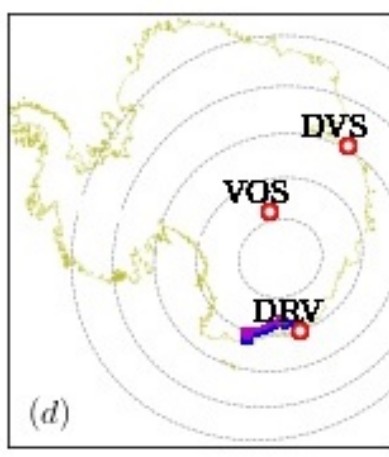

Figure 1: Cluster 1 and 3 satellites orbits in GSE coordinates (XY – panel (a), XZ –(b)) and their projection on the map with observatories located in the (c) Northern and (d) Southern hemispheres (12-18 UT). The initial point and final point are marked with a circle and a triangle, respectively.

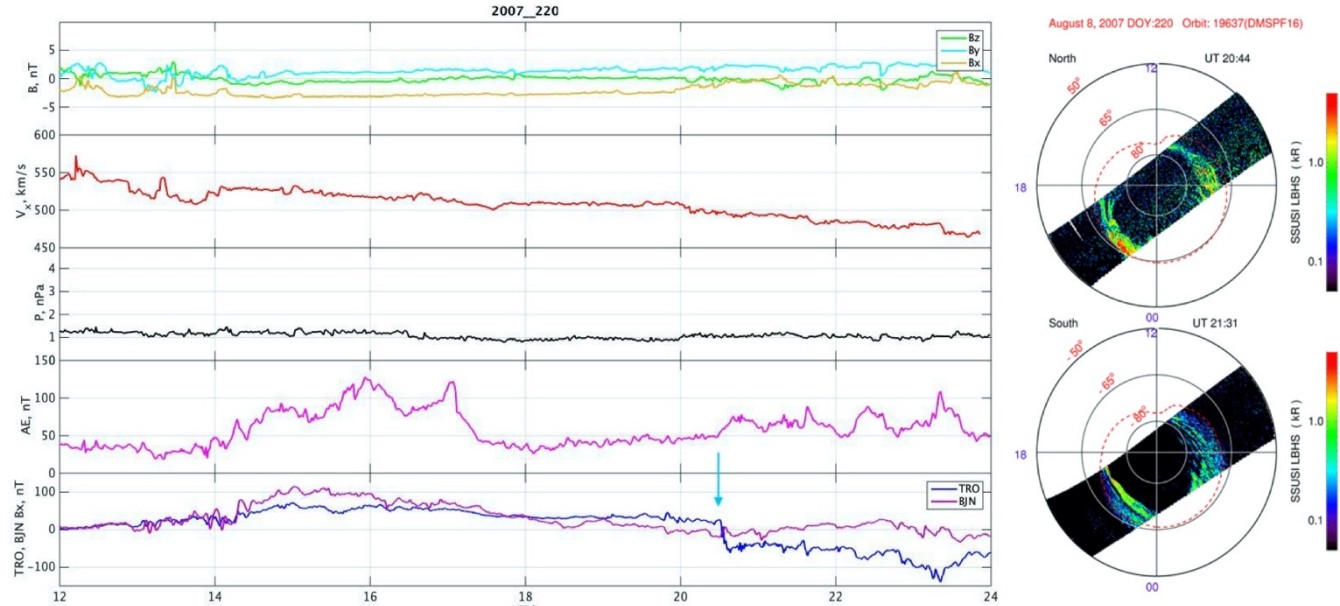

**Figure 2: The particle precipitation seen from the DMSP satellite (right panel) and the solar wind parameters as well as ground based magnetometers data (left panel, from top to bottom: IMF all components; SW speed; SW pressure; AE-index; the magnetic field data from the TRO and BJN stations. The studied substorm is marked with the vertical blue arrow).**

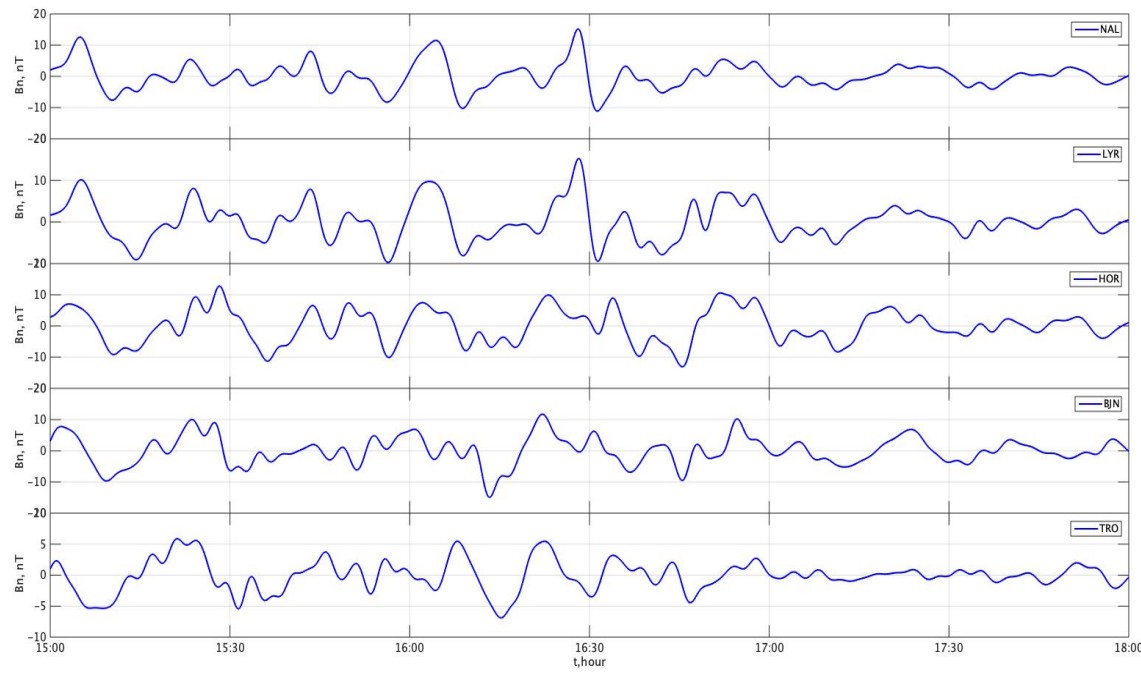

**Figure 3: A latitude profile of geomagnetic pulsations registered on Svalbard (4 top panels) and at Tromsø (the bottom panel).**

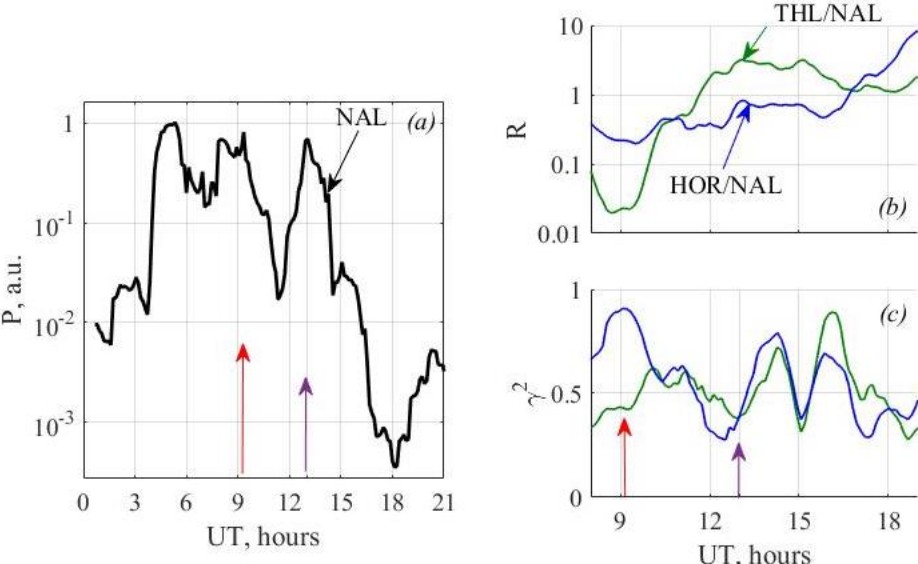

**Figure 4: (a) Diurnal variations of the spectral power in 1.2-1.9 mHz frequency band for NAL $b_N$ component (b) PSD spectral ratio and(c), spectral coherence for the NAL-THL and NAL-HOR station pairs. Near-noon and afternoon maxima are marked by arrows.**

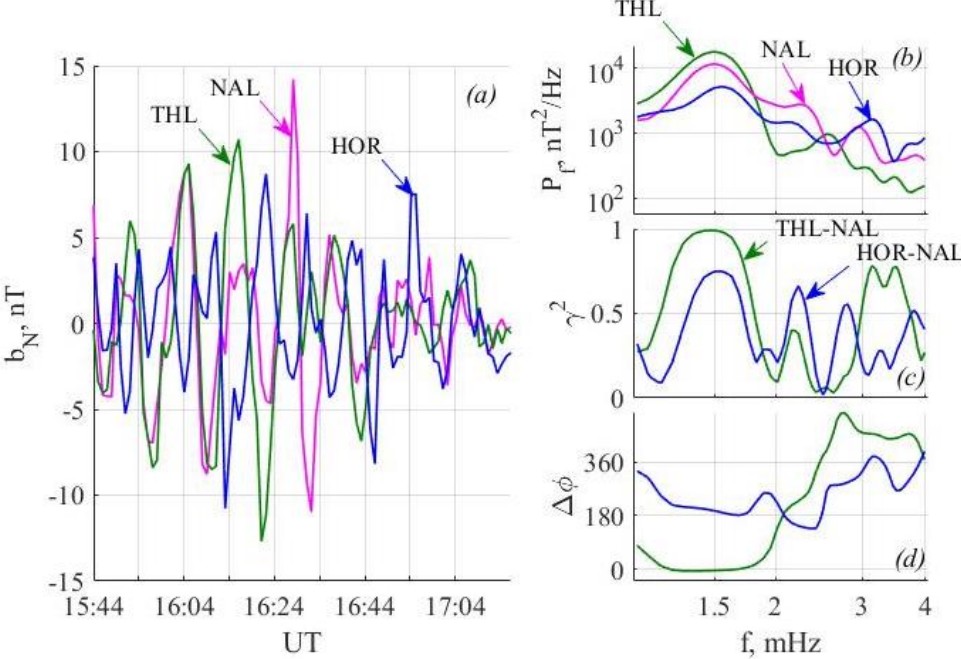

**Figure 5: (a) Pulsations of $b_N$ component at NAL, THL, and HOR during the interval starting at 15:44 UT; (b) PSD spectra; (c) spectral coherence for THL- NAL and HOR- NAL pairs of stations; (d) phase differences. THL- NAL and HOR- NAL station pairs at panels (c) and (d) are shown with the same colours as THL and HOR at panels (a) and (b).**

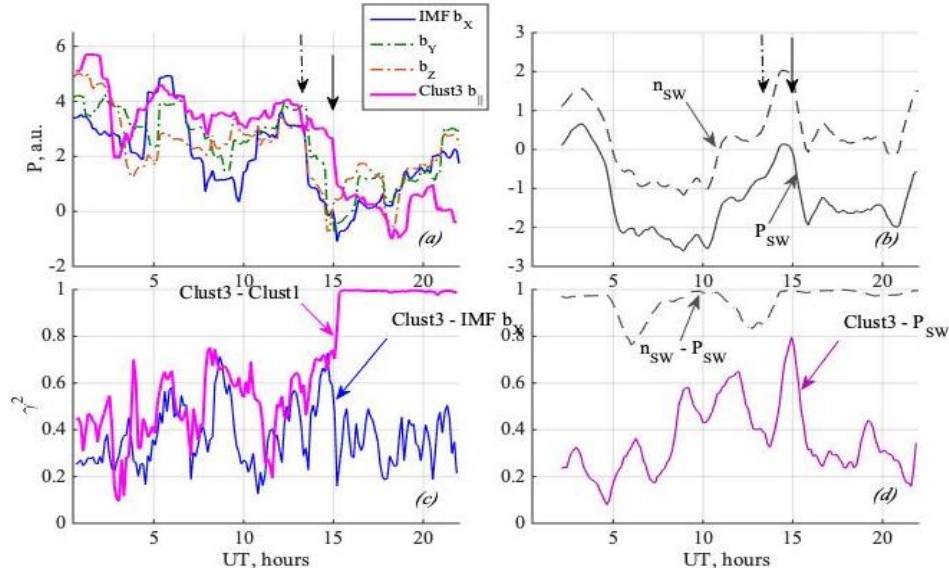

**Figure 6: Left column: Time variations of (a) spectral power and (c) average spectral coherence for the three IMF components (GSM) and field aligned $b_\parallel$ components at Clusters 1 and 3. Right column: (b) variations of total spectral power and (d) average spectral coherence for the SW density and SW dynamic pressure as well as field aligned $b_\parallel$ components at Clusters 3. The dot-dash and solid arrows indicate the time at which there is a significant decrease of the spectral power in the IMF fluctuations (dot-dashed),**

**and in $b_\parallel$ at Cluster and the SW density and pressure (solid).**

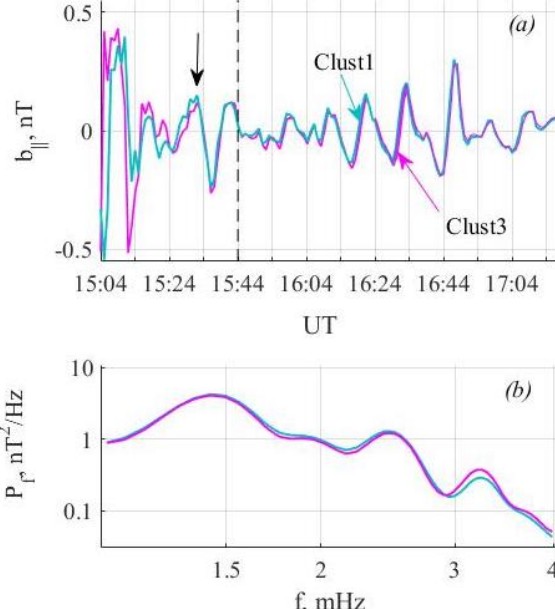

**Figure 7: (a) Pulsations of $b_\parallel$ component in the magnetotail, recorded by Clusters 1 and 3 during the interval starting at 15:04 UT; and (b) PSD spectra for the 96-min interval starting at 15:44 (the start instant of the window, for which the spectra are calculated is marked with a vertical dashed line). The change of pulsation regime is marked with an arrow. Pulsations at Clusters 1 and 3 are**

530 **shown with the same colours in all the panels**

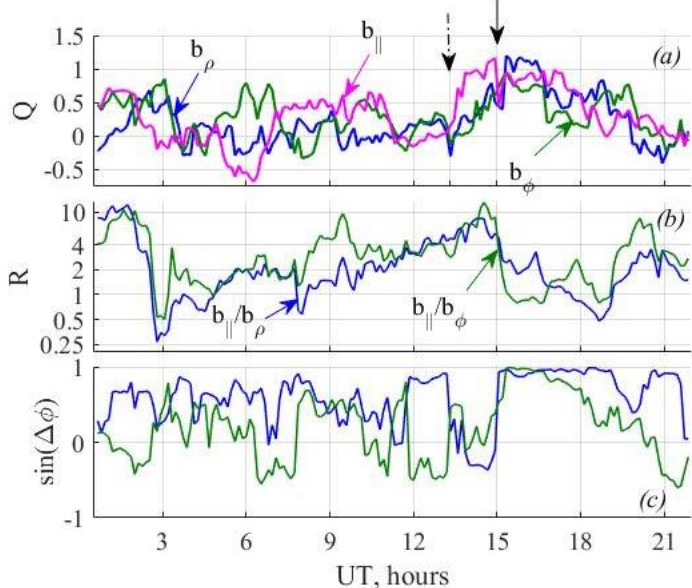

**Figure 8: Time variations of (a) the parameter $Q$, (b) spectral field-aligned to transversal components spectral power ratio $R$, and (c) the sinus of phase difference at Cluster 3. The change of pulsation regime in the IMF (at 13:30UT) and in $b_{\parallel}$ at Cluster at 15 UT is marked by the arrows, similar to Figures 7.**

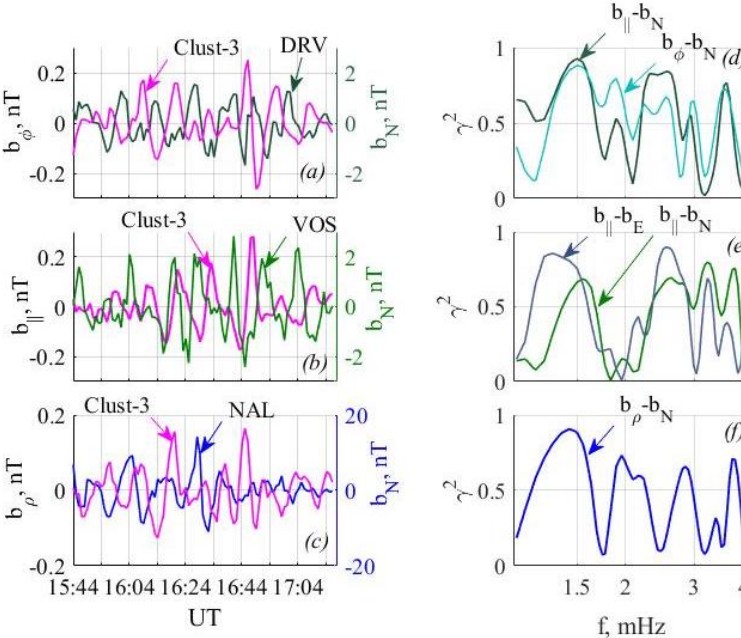

**Figure 9: Pulsations in the magnetotail (Cluster-3) and of $b_N$ on Earth. Pulsations at Cluster 3 and one of the ground stations are shown in panels (a-c). A component with the maximal coherence at the $f_1$ frequency with the corresponding station is shown at each of a-c panels. Left/right Y-axis correspond to Cluster 3/ground, respectively. Spectral coherence for the same satellite –station pairs are given in panels (d-f).**

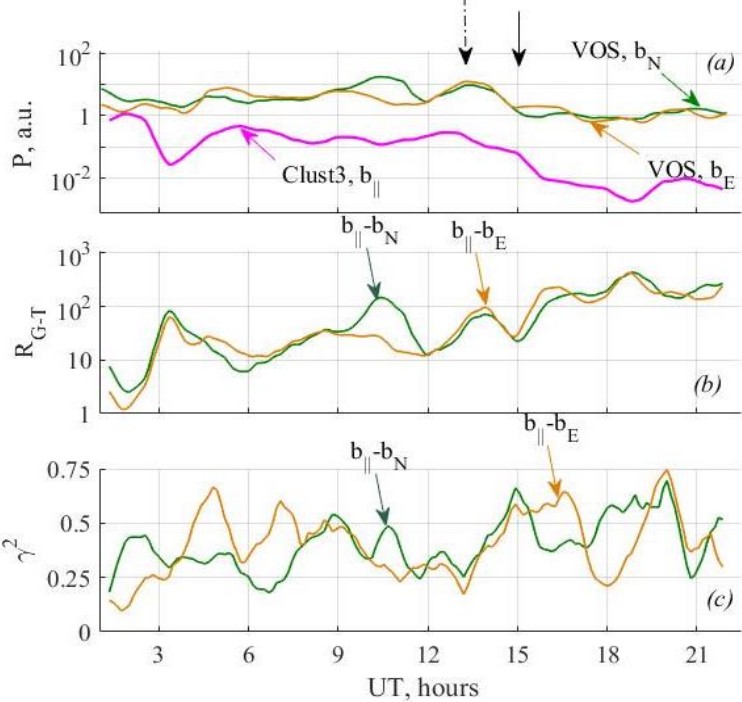

**Figure 10: Time variations of (a) spectral power, (b) ground to tail spectral ratio $R_{G-T}$, and average (c) spectral coherence in the 1.2-1.9 mHz frequency band for $b_\parallel$ components at Cluster 3 and both horizontal components at VOS. Time instants of decrease of spectral power of IMF fluctuations and pulsations in $b_\parallel$ at Cluster are marked with dot-dash and solid arrows, respectively.**

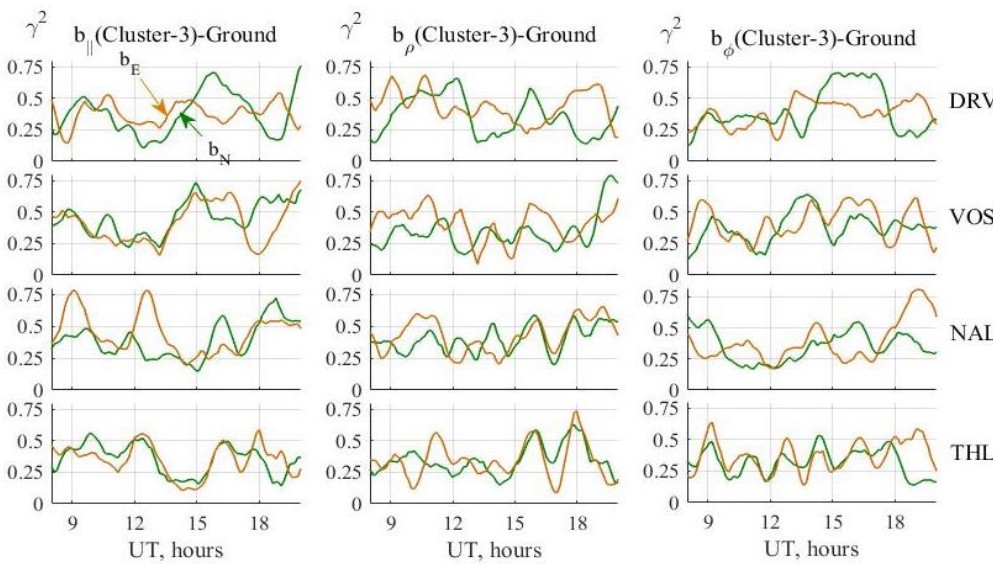

**Figure 11: Time variations of average spectral coherence in 1.2-1.9 mHz frequency band for all component pairs formed from Cluster-3 and the 4 stations. 3 columns correspond to $b_\parallel$, $b_\rho$, and $b_\varphi$ Cluster components, and the rows show the DRV, VOS, NAL, and THL stations. The $b_N$ and $b_E$ components on Earth are shown in green and orange, respectively.**

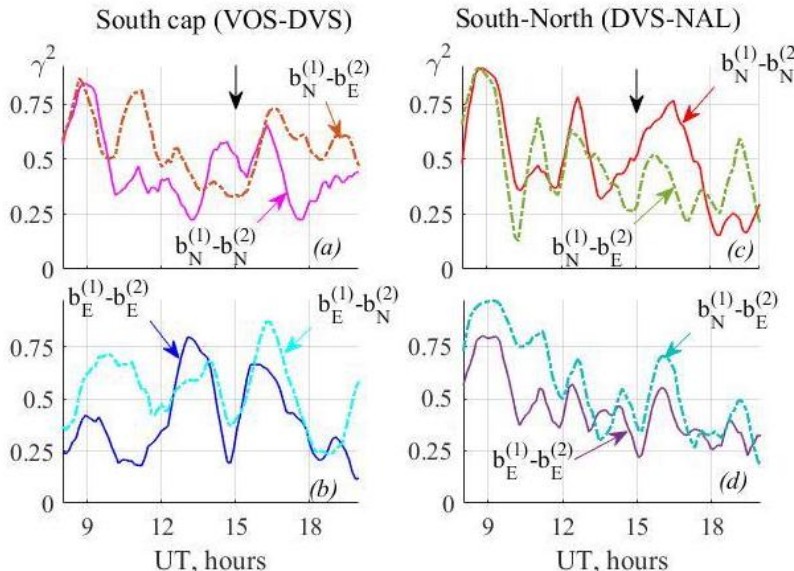

**Figure 12: Time variations of average spectral coherence in 1.2-1.9 mHz frequency band between two points in the Southern polar cap and between the Southern and the Northern polar cap stations: (a) $b_N$ component at VOS - both horizontal components at DVS; (b) $b_E$ component at VOS - both horizontal components at DVS; (c) $b_N$ component at DVS - both horizontal components at NAL; (d) $b_E$ component at DVS - both horizontal components at NAL. Coherence for corresponding components (NN and EE) is shown in all the panels with solid lines, and cross-component (NE and EN) coherence is shown with dash-dot lines. Upper indexes indicate the number of a station in the station pair. The time at which there is a decrease in the spectral power of the pulsations at Cluster are marked with an arrow.**

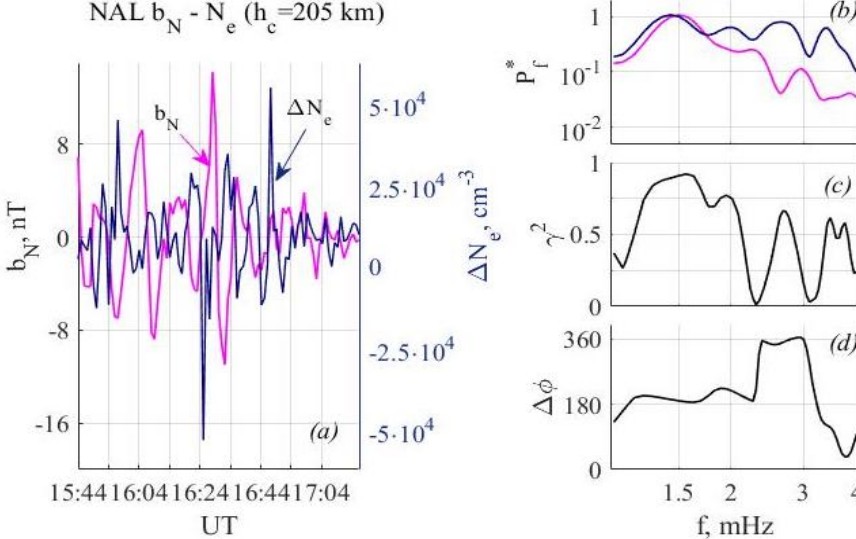

**Figure 13: (a) Pulsations of $b_N$ component at NAL and fluctuations of $N_e$ in the altitude band 190-220 km during the interval starting at 15:44 UT; (b) PSD spectra; (c) spectral coherence; (d) phase differences.**

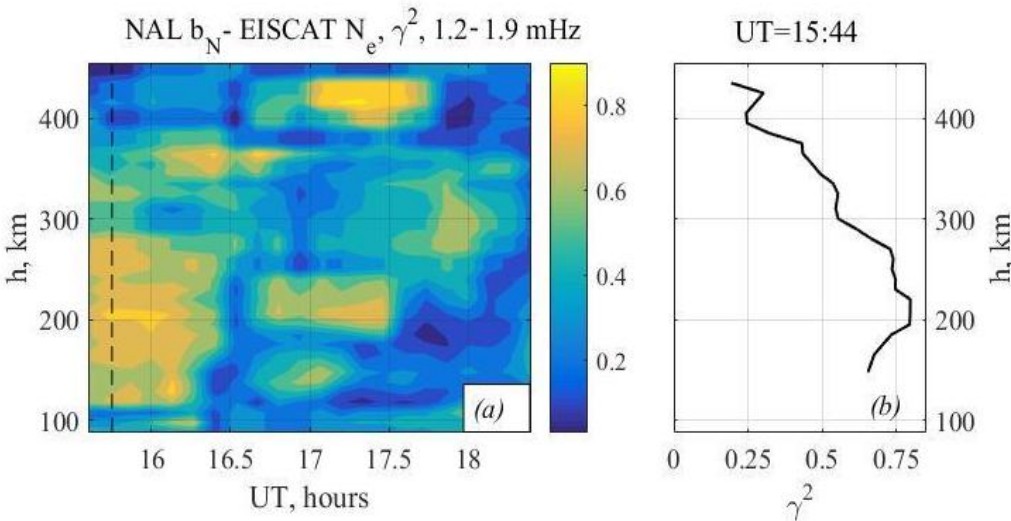

**Figure 14: Dynamic altitude distribution of spectral coherence between EISCAT $N_e$ and NAL $b_N$ in 1.2-1.9 mHz frequency band at evening hours (a), and the altitude distribution of spectral coherence for the 15:44 interval.**