# Peer review of "A case study of the spectral parameters of ULF fluctuations before substorms with no evident trigger in the interplanetary space"

_Annales Geophysicae, 2020_

## Referee Comment (RC1) · Jonathan Rae (Referee) · 9 Nov 2020

This manuscript is a case study of ultra-low frequency (ULF) waves that occur in the magnetosphere and ionosphere before an isolated substorm. These ULF waves are proposed to be either from the solar wind, or from processes inside of the magnetosphere. If these waves are generated by either the solar wind or generated inside of the magnetosphere, then the proposition is that these ULF waves may be considered as a a pre-conditioning of the magnetosphere or trigger for the substorm.

[Figure]

If I understand correctly, the solar wind is ruled out as a generation mechanism, and hence these waves are generated inside of the magnetosphere and these waves could play a role in substorm onset physics.

Most of my concerns are on the conclusions, which are that Pc5 pulsations were observed several hours before the substorm and these pulsations could be related to substorm onset. The substorm starts at 2030 UT. The Pc5 waves occur between ~1500-1800 UT. I'm sorry to say that I can't see the causal link between ULF pulsations that occur 2-3 hours before a small substorm bay and the manuscript does not say how this link really works. I found the analysis of the Pc5 waves to be a very nice study of their coherence, but I'm sorry to say that I don't see the relationship between these waves and a substorm that happens 2-3 hours later. For example, the average substorm cycle is around 2-3 hours, and the average substorm expansion phase is ~minutes, and so it is not clear to me how these waves and this substorm are related when they are observed hours apart.

I do have a suggestion, which is that I think that the Pc5 wave analysis is very good and very clear. It is possible that this analysis and interpretation could form a manuscript all by itself without the link to substorms. Whether that is something that the authors were intending and I misunderstood I do not know but if I did misunderstand then I apologise. In summary, I would suggest either the manuscript is clearly rewritten to make clear how the Pc5 waves are related to the substorm, or to just concentrate on the very nice analysis of the Pc5 waves themselves and not relate them to the substorm onset process which is not quite demonstrated at this point. At the moment the manuscript seems to be somewhere in between these two points of view.

A few more comments on the general introduction and discussion. I would recommend that the referencing of the relevant publications in the field could be added to. There are plenty of references on ULF waves as a trigger for substorm onset, originating with the seminal work of Samson and most of those references could be found within Rae et al. [2014] and Smith et al. [2019] - that itself cites a wide range of Samson

and other relevant papers with which the literature review could be improved. This is also true in the Introduction, where there are very few substorm references to be found (lines 20-30) other than the Bland et al. reference. Some of these papers might not be relevant but I offer these as papers who discuss ULF waves, Pc5 waves and potential triggering of substorm onset. I'm happy to provide more references to the authors if that is helpful.

I would recommend that the introduction could be expanded to include some of the literature that discusses the physics of triggering of substorm onset by the solar wind and by internal processes.

Data Processing. I would recommend that the data processing aspects of the work are discussed in the sections closer to the Figures that are being described. I think that it would aid in the readability of the manuscript if the data processing were to be close to the Figures being described.

Figures. I would recommend using Universal Time for the figures, instead of "time since 1530 UT". It would really help the readability of the paper to use a common time for each plot so that the reader can understand where each plot starts and finishes relative to the others.

References Rae, I. J., Murphy, K. R., Watt, C. E. J., Rostoker, G., Rankin, R., Mann, I. R., Hodgson, C. R., Frey, H. U., Degeling, A. W., and Forsyth, C. (2014), Field line resonances as a trigger and a tracer for substorm onset, J. Geophys. Res. Space Physics, 119, 5343– 5363, doi:10.1002/2013JA018889. Samson, J. C., D. D. Wallis, T. J. Hughes, F. Creutzberg, J. M. Ruohoniemi, and R. A. Greenwald (1992), Substorm intensifications and field line resonances in the nightside magnetosphere, J. Geophys. Res., 97, 8495–8518, doi:10.1029/91JA03156. Smith, A. W., Rae, I. J., Forsyth, C., Watt, C. E. J., & Murphy, K. R. (2020). On the magnetospheric ULF wave counterpart of substorm onset. Journal of Geophysical Research: Space Physics, 125, e2019JA027573. https://doi.org/10.1029/2019JA027573

---

## Author Comment (AC1) · 23 Nov 2020

This manuscript is a case study of ultra-low frequency (ULF) waves that occur in the magnetosphere and ionosphere before an isolated substorm. These ULF waves are proposed to be either from the solar wind, or from processes inside of the magnetosphere. If these waves are generated by either the solar wind or generated inside of the magnetosphere, then the proposition is that these ULF waves may be considered as a a pre-conditioning of the magnetosphere or trigger for the substorm.

If I understand correctly, the solar wind is ruled out as a generation mechanism, and hence these waves are generated inside of the magnetosphere and these waves could play a role in substorm onset physics.

Most of my concerns are on the conclusions, which are that Pc5 pulsations were observed several hours before the substorm and these pulsations could be related to substorm onset. The substorm starts at 2030 UT. The Pc5 waves occur between ∼1500-1800 UT. I'm sorry to say that I can't see the causal link between ULF pulsations that occur 2-3 hours before a small substorm bay and the manuscript does not say how this link really works. I found the analysis of the Pc5 waves to be a very nice study of their coherence, but I'm sorry to say that I don't see the relationship between these waves and a substorm that happens 2-3 hours later. For example, the average substorm cycle is around 2-3 hours, and the average substorm expansion phase is ∼minutes, and so it is not clear to me how these waves and this substorm are related when they are observed hours apart.

This manuscript is based on the previous paper (Yagova et al, 2017) and it develops the idea proposed in the paper. The mentioned study is aimed on the investigation of

geomagnetic and auroral luminosity pulsations in the frequency range 1-4 mHz using ground-based magnetometer and Meridional Scanning Photometer data. Days with the undisturbed solar wind and interplanetary magnetic field parameters and were selected and divided into 2 sets: days with a substorm and quiet days. The second set was used to determine a background variation of the spectral parameters of the ULF pulsations. The first set consists of 15 non-triggered substorms, i.e. substorms without any evident trigger in the IMF nor SW; 7 out of 15 substorms were considered as an isolated, i.e. separated from other substorm with at list 3 hours. It was shown, that for days with substorms, the PSD of Pc5/Pi3 geomagnetic pulsations, recorded in the Northern Polar Cap for several hours, preceding a substorm, is much higher as compared with the quiet days. Such pulsations are coherent with fluctuations in auroral luminosity (557.7 nm). The analysis also showed distinct spectral changes (as determined by the spectral slope α and the parameter Q) during 3-4 hours before a substorm onset. It was speculated that these pulsations might be caused by ULF activity in the IMF and SW or be a «precursor» of an upcoming substorm. If the first scenario is realized, then ULF fluctuations should be added in the list of substorm triggers. In the second scenario the pulsations could indicate a preparation of the magnetosphere to a substorm.

The present case study illustrates the possibility of the second scenario by combining ground magnetometer with in-situ measurements inside the magnetosphere (from Cluster). The analysis is thus focused on the time period, 3-4 hours before substorm onset, in line with the finding from Yagova et al (2017). While ULF fluctuations almost die out in the SW and IMF, ULF activity starts to grow in the Magnetosphere as shown from both the Cluster and ground magnetometer data.

In the revised version we plan to include the main results of Yagova et al., 2017 and the preceding research on the relationship between an auroral substorm and ULF activity in the polar caps in the Introduction section. We will also discuss the possible physical reasons in the Discussion section.

I do have a suggestion, which is that I think that the Pc5 wave analysis is very good and very clear. It is possible that this analysis and interpretation could form a manuscript all by itself without the link to substorms. Whether that is something that the authors were intending and I misunderstood I do not know but if I did misunderstand then I apologise. In summary, I would suggest either the manuscript is clearly rewritten to make clear how the Pc5 waves are related to the substorm, or to just concentrate on the very nice analysis of the Pc5 waves themselves and not relate them to the substorm onset process which is not quite demonstrated at this point. At the moment the manuscript seems to be somewhere in between these two points of view.

We will add more detailed description of the previous study to make the connection between Pc5 waves observed in the 3-4 hours preceding a substorm more evident to the reader.

A few more comments on the general introduction and discussion. I would recommend that the referencing of the relevant publications in the field could be added to. There are plenty of references on ULF waves as a trigger for substorm onset, originating with the seminal work of Samson and most of those references could be found within Rae et al. [2014] and Smith et al. [2019] - that itself cites a wide range of Samson and other relevant papers with which the literature review could be improved. This is also true in the Introduction, where there are very few substorm references to be found (lines 20-30) other than the Bland et al. reference. Some of these papers might not be relevant but I offer these as papers who discuss ULF waves, Pc5 waves and potential triggering of substorm onset. I'm happy to provide more references to the authors if that is helpful.

Thank you so much for the provided references. Two of them were cited in the previous paper. It also was mentioned there, that most of studies are focused on the last pre-substorm minutes and just a few of them pay attention to the longterm preceding period. We plan to discuss the results of these publications and to add the newest suggested paper in the introduction of the current manuscript.

I would recommend that the introduction could be expanded to include some of the literature that discusses the physics of triggering of substorm onset by the solar wind and by internal processes.

We will expand the introduction with the recommended topic.

Data Processing. I would recommend that the data processing aspects of the work are discussed in the sections closer to the Figures that are being described. I think that it would aid in the readability of the manuscript if the data processing were to be close to the Figures being described.

We will give a short data processing description in each section

Figures. I would recommend using Universal Time for the figures, instead of "time since 1530 UT". It would really help the readability of the paper to use a common time for each plot so that the reader can understand where each plot starts and finishes relative to the others.

We will replot figures with a common time axis.

References Rae, I. J., Murphy, K. R., Watt, C. E. J., Rostoker, G., Rankin, R., Mann, I. R., Hodgson, C. R., Frey, H. U., Degeling, A. W., and Forsyth, C. (2014), Field line resonances as a trigger and a tracer for substorm onset, J. Geophys. Res. Space Physics, 119, 5343– 5363, doi:10.1002/2013JA018889. Samson, J. C., D. D. Wallis, T. J. Hughes, F. Creutzberg, J. M. Ruohoniemi, and R. A. Greenwald (1992), Substorm intensifications and field line resonances in the nightside magnetosphere, J. Geophys. Res., 97, 8495–8518, doi:10.1029/91JA03156. Smith,

A. W., Rae, I. J., Forsyth, C., Watt, C. E. J., & Murphy, K. R. (2020). On the magnetospheric ULF wave coun- terpart of substorm onset. Journal of Geophysical Research: Space Physics, 125, e2019JA027573. https://doi.org/ 10.1029/2019JA027573

---

## Referee Comment (RC2) · Emil Kepko (Referee) · 30 Nov 2020

This paper presents a case study of ULF oscillations observed in the magnetotail by Cluster, on the ground with ground magnetometers, and in ionospheric density. The paper demonstrates coherence between the different magnetospheric regions. As with the other reviewer, I find the connection (or not) with substorms tenuous, at best. Conclusions 1 & 2 are conjecture but are not supported by the data in the paper. I suggest simply removing that aspect, so I will not repeat those concerns here – but I fully concur with the other reviewer.

[Figure]

I have some questions regarding the spectral analysis and data presentation. With respect to external vs. internal driving of the oscillations, I would encourage the authors to examine the solar wind number density, in addition to the IMF, as the solar wind dynamic pressure – whose variations are driven primarily by number density variations – is the largest driver of ULF pulsations in this frequency range. The fact that these pulsations were observed in the polar cap certainly limits the internal mechanisms.

I also would like to see a bit more information on how the spectra were computed. The paper mentioned a low-pass filter – what was the filter type? It is important to present those details, as certain techniques (e.g., running average) can introduce spurious spectral signals. The spectra look very smooth, so I'm curious as well how those were computed. The paper states Blackman-Tukey but, specifically, what is the effective Rayleigh frequency of the analysis? The spectra look very smooth. Finally, it is difficult to judge the purported peak near 1.5 mHz without seeing the lower frequencies. I Suggest plotting to the zero frequency so a reader can see the context for this peak. Relative to the background, the peak near 1.5 looks little different than other peaks (e.g., near 3 mHz). Plotting to lower frequencies would help the reader be able to determine significance, at least visually. There are statistical tests one could apply as well (see our recent Kepko, Viall & Wolfinger paper for one, but there are many others), and I would encourage looking at those, although the cross-coherence is an indicator that these are geophysical phenomena.

---

## Author Comment (AC2) · 24 Dec 2020

This paper presents a case study of ULF oscillations observed in the magnetotail by Cluster, on the ground with ground magnetometers, and in ionospheric density. The paper demonstrates coherence between the different magnetospheric regions. As with the other reviewer, I find the connection (or not) with substorms tenuous, at best. Conclusions 1 & 2 are conjecture but are not supported by the data in the paper. I suggest simply removing that aspect, so I will not repeat those concerns here – but I fully concur with the other reviewer.

We understand the problem, that a possible relation between ULF in the Polar cap and a forthcoming substorm cannot be proved with the analysis of a single event. In any case, it was not the purpose of the present study. The existence of a statistical relation between ULF in the polar cap and auroral activation was found statistically by (Heacoc and Chao, 1980, https://doi.org/10.1029/JA085iA03p01203 and Yagova et al, 2000, http://adsabs.harvard.edu/full/2000ESASP.443..603Y). However, these papers did not discriminate between the externally and internally triggered substorms.

This step was done in our previous paper (Yagova et al, 2017), where the analysis of ULF in the polar cap was undertaken intentionally for non-triggered substorms. This analysis led to the conclusion of two possible reasons of pre-substorm activations in the polar caps, the first related to increased level of ULF fluctuations in the IMF and SW plasma parameters, and the other links to inter-magnetospheric ULF waves.

The present study gives a detailed analysis of ULF waves in the magnetosphere and on the ground developing under undisturbed IMF/SW. This, on the one hand, shows a possibility of the second scenario and, on the other hand, allows us to formulate a hypothesis about properties of these waves for future studies.

In the revised version of the paper, we plan to extend the Introduction and Discussion sections to clarify the relevance of the present research in the problem of substorm triggering and, in particular, of ULF activity in the polar caps and the magnetotail.

I have some questions regarding the spectral analysis and data presentation. With respect to external vs. internal driving of the oscillations, I would encourage the authors to examine the solar wind number density, in addition to the IMF, as the solar wind dynamic pressure – whose variations are driven primarily by number density variations – is the largest driver of ULF pulsations in this frequency range. The fact that these pulsations were observed in the polar cap certainly limits the internal mechanisms.

In the revised version we will include an analysis of the solar wind dynamic pressure and number density fluctuations. The physical mechanisms for the MHD wave excitations on the inhomogeneities on open filed lines will be also discussed in the revised version of the MS (see, e.g. Pilipenko et al., 2005, https://doi.org/10.1029/2004JA010755)

I also would like to see a bit more information on how the spectra were computed. The paper mentioned a low-pass filter – what was the filter type? It is important to present those details, as certain techniques (e.g., running average) can introduce spurious spectral signals.

The Data Analysis section will be extended, and all the details of filters and spectral estimates will be given. We understand the problem of the artificial peaks originated from the non-adequate filtering. To exclude any uncertainties, we've plotted the non-filtered signals with removed trends.

[Figure]

Fig.1. Signals with removed trends seen in (from top to bottom): the magnetotail in Cluster-3 transversal components; IMF By component.

The spectra look very smooth, so I'm curious as well how those were computed. The paper states Blackman-Tukey but, specifically, what is the effective Rayleigh frequency of the analysis? The spectra look very smooth. Finally, it is difficult to judge the purported peak near 1.5 mHz without seeing the lower frequencies. I Suggest plotting to the zero frequency so a reader can see the context for this peak. Relative to the background, the peak near 1.5 looks little different than other peaks (e.g., near 3 mHz). Plotting to lower frequencies would help the reader be able to determine significance, at least visually.

The Blackman-Tukey method is well known and widely used in geophysics. In contrast to methods such as maximal Entropy (MEM), it has a worth frequency resolution, but it estimates PSD with a dispersion which decreases with spectral smoothing. In fact, the parameters were chosen as a compromise between the frequency resolution and dispersion of spectral estimates. The spectra of non-filtered signals are given in Figure 2. It can be seen from the Figure that the main spectral maximum at 1.5 mHz is not a result of filtering. Besides, the nearly 11-minute pulsations are clearly seen in time series, as well (fig.1).

In the revised version of the manuscript, we will give a description of the technique in more detail.

[Figure]

Fig.2. PSD spectra for (from top to bottom): NAL ground-based magnetic data; Cluster 3 field aligned component.

There are statistical tests one could apply as well (see our recent Kepko, Viall & Wolfinger paper for one, but there are many others), and I would encourage looking at those, although the cross-coherence is an indicator that these are geophysical phenomena.

Thank you for the method suggested. We will consider the possibility of applying this technique to our future studies.

---

## Author Response (AR1)

**Jonathan Rae (Referee)**  jonathan.rae@unorthumbria.ac.uk

This manuscript is a case study of ultra-low frequency (ULF) waves that occur in the magnetosphere and ionosphere before an isolated substorm. These ULF waves are proposed to be either from the solar wind, or from processes inside of the magnetosphere. If these waves are generated by either the solar wind or generated inside of the magnetosphere, then the proposition is that these ULF waves may be considered as a a pre-conditioning of the magnetosphere or trigger for the substorm.

If I understand correctly, the solar wind is ruled out as a generation mechanism, and hence these waves are generated inside of the magnetosphere and these waves could play a role in substorm onset physics. Most of my concerns are on the conclusions, which are that Pc5 pulsations were observed several hours before the substorm and these pulsations could be related to substorm onset. The substorm starts at 2030 UT. The Pc5 waves occur between ~1500-1800 UT. I'm sorry to say that I can't see the causal link between ULF pulsations that occur 2-3 hours before a small substorm bay and the manuscript does not say how this link really works. I found the analysis of the Pc5 waves to be a very nice study of their coherence, but I'm sorry to say that I don't see the relationship between these waves and a substorm that happens 2-3 hours later. For example, the average substorm cycle is around 2-3 hours, and the average substorm expansion phase is ~minutes, and so it is not clear to me how these waves and this substorm are related when they are observed hours apart.

This manuscript is based on the previous paper (Yagova et al, 2017) and it develops the idea proposed in the paper. In the revised version we have included the main results of Yagova et al., 2017 and the preceding papers about the relationship between an auroral substorm and ULF activity in the polar caps in the Introduction section (55-93). A brief discussion of a longer preparation phase of a substorm developing at a non-disturbed SW/IMF is given in the Discussion section (395-399).

I do have a suggestion, which is that I think that the Pc5 wave analysis is very good and very clear. It is possible that this analysis and interpretation could form a manuscript all by itself without the link to substorms. Whether that is something that the authors were intending and I misunderstood I do not know but if I did misunderstand then I apologise. In summary, I would suggest either the manuscript is clearly rewritten to make clear how the Pc5 waves are related to the substorm, or to just concentrate on the very nice analysis of the Pc5 waves themselves and not relate them to the substorm onset process which is not quite demonstrated at this point. At the moment the manuscript seems to be somewhere in between these two points of view.

In the revised version a more detailed description of the previous study is given in order to make the connection between Pc5 waves observed in the 3-4 hours preceding a substorm more evident to the reader (74-93).

A few more comments on the general introduction and discussion. I would recommend that the referencing of the relevant publications in the field could be added to. There are plenty of references on ULF waves as a trigger for substorm onset, originating with the seminal work of Samson and most of those references could be found within Rae et al. [2014] and Smith et al. [2019] - that itself cites a wide range of Samson and other relevant papers with which the literature review could be improved. This is also true in the Introduction, where there are very few substorm references to be found (lines 20-30) other than the Bland et al. reference. Some of these papers might not be relevant but I offer these as papers who discuss ULF waves, Pc5 waves and potential triggering of substorm onset. I'm happy to provide more references to the authors if that is helpful.

In the revised version the role of Alfven FLR in substorm triggering in discussed in more details and the references suggested are included into review (55-62).

I would recommend that the introduction could be expanded to include some of the literature that discusses the physics of triggering of substorm onset by the solar wind and by internal processes.

The Introduction is extended, and the problem of external and internal processes are discussed (28-57).

Data Processing. I would recommend that the data processing aspects of the work are discussed in the sections closer to the Figures that are being described. I think that it would aid in the readability of the manuscript if the data processing were to be close to the Figures being described.

Now, a short data processing description is given in each section (165-170, 176-177, 245-250)

Figures. I would recommend using Universal Time for the figures, instead of "time since 1530 UT". It would really help the readability of the paper to use a common time for each plot so that the reader can understand where each plot starts and finishes relative to the others.

The format of time is changed to make it common for all the figures.

This paper presents a case study of ULF oscillations observed in the magnetotail by Cluster, on the ground with ground magnetometers, and in ionospheric density. The paper demonstrates coherence between the different magnetospheric regions. As with the other reviewer, I find the connection (or not) with substorms tenuous, at best. Conclusions 1 & 2 are conjecture but are not supported by the data in the paper. I suggest simply removing that aspect, so I will not repeat those concerns here – but I fully concur with the other reviewer.

We understand the problem, that a possible relation between ULF in the Polar cap and a forthcoming substorm cannot be proved with the analysis of a single event. In any case, it was not the purpose of the present study. This problem was studied statistically which were published in several papers. Now, we have extended the Introduction and Discussion sections to clarify the relevance of the present research in the problem of substorm triggering and, in particular, of ULF activity in the polar caps and the magnetotail (27-93, 362-373, 395-399).

I have some questions regarding the spectral analysis and data presentation. With respect to external vs. internal driving of the oscillations, I would encourage the authors to examine the solar wind number density, in addition to the IMF, as the solar wind dynamic pressure – whose variations are driven primarily by number density variations – is the largest driver of ULF pulsations in this frequency range. The fact that these pulsations were observed in the polar cap certainly limits the internal mechanisms.

In the revised version we have included an analysis of the solar wind dynamic pressure and number density fluctuations (Fig.6, lines 229-239). The physical mechanisms for the MHD wave excitations on the inhomogeneities on open filed lines is discussed in the revised version of the MS (70-72,392-394).

I also would like to see a bit more information on how the spectra were computed. The paper mentioned a low-pass filter – what was the filter type? It is important to present those details, as certain techniques (e.g., running average) can introduce spurious spectral signals.

The Data Analysis section is extended in order to clarify the details of filters and spectral estimates (138-142). We understand the problem of the artificial peaks originated from the non-adequate filtering. To exclude any uncertainties, a supplementary file with spectra of a non-filtered signal can be found in the Figure S1 in the supplementary materials.

The spectra look very smooth, so I'm curious as well how those were computed. The paper states Blackman-Tukey but, specifically, what is the effective Rayleigh frequency of the analysis? The spectra look very smooth. Finally, it is difficult to judge the purported peak near 1.5 mHz without seeing the lower frequencies. I Suggest plotting to the zero frequency so a reader can see the context for this peak. Relative to the background, the peak near 1.5 looks little different than other peaks (e.g., near 3 mHz). Plotting to lower frequencies would help the reader be able to determine significance, at least visually.

In the revised version of the manuscript, a description of the spectral analysis is given in more detail (122-124).

There are statistical tests one could apply as well (see our recent Kepko, Viall & Wolfinger paper for one, but there are many others), and I would encourage looking at those, although the cross-coherence is an indicator that these are geophysical phenomena.

Thank you for the method suggested. We have added the citation in the Introduction (47-50) and we will consider the possibility of applying this technique to our future studies.

---

## Author Response (AR2)

Dear Br.Rae!

Thank you so much for your report and all comments. We are really sorry that our arguments do not sound convincingly enough. Taken this in account, we've changed the draft according to your suggestions. All changes are marked with colour.

We totally agree, that it is not possible to prove with the used data set, that the studied pulsations are related to the substorm. But observations of special pre-substorm ULF fluctuations in the polar cap have been reported in a few papers, including our previous one (Yagova et al, 2017). It was statistically shown, that Pc5/Pi3 pulsations with some distinguishing features are seen in the Polar cap for days with a substorm and are absent in quiet days. The present event perfectly matches the parameters used in the mentioned paper to identify a non-triggered isolated substorm. That is why we expected to see pulsations in the Polar cap for the studied day and we did find them. This allows us to speculate about a possible relationship between the pulsations and the substorm. In future studies we will try to investigate this idea with extended sets of data. As for now, we follow all your recommendations and appreciate you for your careful reading and useful comments.

sincerely,
on behalf of authors,
Nataliya Nosikova